Prokaryotic communities from a lava tube cave in La Palma Island (Spain) are involved in the biogeochemical cycle of major elements

Gonzalez-Pimentel Jose Luis 1
Martin-Pozas Tamara 2
Jurado Valme 3
Miller Ana Zelia 1
Caldeira Ana Teresa 1
Fernandez-Lorenzo Octavio 4
Sanchez-Moral Sergio 2
Saiz-Jimenez Cesareo saiz@irnase.csic.es 3
1 Laboratorio Hercules, Universidade de Evora , Evora , Portugal
2 Geology, Museo Nacional de Ciencias Naturales, CSIC , Madrid , Spain
3 Environmental Microbiology, Instituto de Recursos Naturales y Agrobiologia, CSIC , Sevilla , Spain
4 Grupo de Espeleologia Tebexcorade , La Palma , Spain
Uversky Vladimir
Electronic publication date: 2021 May 11
Publication date: 2021
Volume: 9
Electronic Location ID: e11386
Received 2021 Feb 26; Accepted 2021 Apr 10
Copyright: ©2021 Gonzalez-Pimentel et al.
Copyright year: 2021
Copyright holder: Gonzalez-Pimentel et al.
License: This is an open access article distributed under the terms of the Creative Commons Attribution License, which permits unrestricted use, distribution, reproduction and adaptation in any medium and for any purpose provided that it is properly attributed. For attribution, the original author(s), title, publication source (PeerJ) and either DOI or URL of the article must be cited.
License URL: https://creativecommons.org/licenses/by/4.0/

Keywords: Lava tube, Volcanic cave, La Palma Island, Biogeochemical cycles, Proteobacteria

Funding: Spanish Ministry of Economy and Competitiveness CGL2013-41674-P CGL2016-75590-P PID2019-110603RB-I00 AEI/FEDER UE and CSIC Open Access Publication Support Initiative This work was supported by the Spanish Ministry of Economy and Competitiveness through projects CGL2013-41674-P, CGL2016-75590-P, PID2019-110603RB-I00, AEI/FEDER, UE and CSIC Open Access Publication Support Initiative through its Unit of Information Resources for Research (URICI). The funders had no role in study design, data collection and analysis, decision to publish, or preparation of the manuscript.

==============================
Lava caves differ from karstic caves in their genesis and mineral composition. Subsurface microbiology of lava tube caves in Canary Islands, a volcanic archipelago in the Atlantic Ocean, is largely unknown. We have focused the investigation in a representative lava tube cave, Fuente de la Canaria Cave, in La Palma Island, Spain, which presents different types of speleothems and colored microbial mats. Four samples collected in this cave were studied using DNA next-generation sequencing and field emission scanning electron microscopy for bacterial identification, functional profiling, and morphological characterization. The data showed an almost exclusive dominance of Bacteria over Archaea. The distribution in phyla revealed a majority abundance of Proteobacteria (37–89%), followed by Actinobacteria, Acidobacteria and Candidatus Rokubacteria. These four phyla comprised a total relative abundance of 72–96%. The main ecological functions in the microbial communities were chemoheterotrophy, methanotrophy, sulfur and nitrogen metabolisms, and CO2 fixation; although other ecological functions were outlined. Genome annotations of the especially representative taxon Ga0077536 (about 71% of abundance in moonmilk) predicted the presence of genes involved in CO2 fixation, formaldehyde consumption, sulfur and nitrogen metabolisms, and microbially-induced carbonate precipitation. The detection of several putative lineages associated with C, N, S, Fe and Mn indicates that Fuente de la Canaria Cave basalts are colonized by metabolically diverse prokaryotic communities involved in the biogeochemical cycling of major elements.

Introduction

Lava tube caves are formed as a result of surface solidification of a lava flow during the last stages of volcanic activity. A decreasing supply of lava may then cause the molten material to drain out from under this crust and leave long cylindrical tunnels or caves. In spite of the existence of abundant literature on the tectonism, volcanism and geology of volcanic ocean islands (Carracedo & Troll, 2016; Beier, Haase & Brandl, 2018), the microbiology of lava tube caves was a topic scarcely studied in the first decade of the present century (Northup et al., 2004; Snider et al., 2009; Snider, 2010). However, in this decade lava tube caves become a hot spot for microbiologists, among other reasons by the interest in the search of recognizable biosignatures useful as astrobiology models due to the analogy of Earth volcanic lava caves with those of Mars (Perkins, 2020).

A large number of studies on the microbiology of lava tube caves have been dedicated to Hawai’i (Northup et al., 2011; Hathaway et al., 2014b; Riquelme et al., 2015b; Spilde et al., 2016) and other USA states (Northup et al., 2011; Popa et al., 2012; Riquelme et al., 2015a; Lavoie et al., 2017) and Azores Islands (De los Ríos et al., 2011; Northup et al., 2011; Hathaway et al., 2014a; Hathaway et al., 2014b; Riquelme et al., 2015a; Riquelme et al., 2015b). A few studies were also published on lava tube caves from Mexico, Galápagos and Easter islands (Luis-Vargas et al., 2019; Miller et al., 2014; Miller et al., 2020b).

The Canary Islands, Spain, is a volcanically active archipelago with eleven islands and islets, located in the Macaronesia region. The last eruptions occurred in La Palma (1971) and in El Hierro islands (2011-2012). In spite of the importance of this archipelago, geomicrobiological study of Canary Islands lave tube caves are rare (Gonzalez-Pimentel, 2019; Gonzalez-Pimentel et al., 2018).

The data reported on microbial diversity of lava tube caves in other volcanic islands need to be contrasted with those obtained for the relatively unknown lava tube caves from the Canary Islands, and so we focused the investigation in Fuente de la Canaria Cave, a representative lava tube cave with different types of speleothems and microbial mats. It is expected that the diversity of studied niches in this volcanic cave will shed light on the microbiology of lava tube caves from La Palma Island and will permit a comparison with other previously studied oceanic volcanic caves.

Materials & Methods

Geological context and research background

La Palma Island (28°40′N, 17°52′W) is located in the northwest of the Canary archipelago. The geology of La Palma was extensively described by Carracedo et al. (2001). This island shows a particular topography, characterized by the volcanos Taburiente and Cumbre Vieja which were separated by a valley. Cumbre Vieja is a volcanic ridge formed by numerous volcanic cones built of lava and scoria, developed over the past 150,000 years at the southern part of the island. The volcano is still active and the last eruption occurred in 1971 at the Teneguía vent, at the southern end of the Cumbre Vieja (Carracedo & Troll, 2016).

Fernández Lorenzo (2007) compiled 135 lava tube caves in La Palma. However, in the last years the number increased due to the continuous discovering of new lava tube caves. The lava tube cave studied, Fuente de la Canaria Cave, is located in the Cumbre Vieja volcano area, which is characterized by the predominance of alkaline-dominated basaltic lavas (Klügel, Hansteen & Galipp, 2005).

Miller et al. (2020a) described the climate of La Palma Island as very mild and sunny most of the year, with rainfalls in autumn and winter. Humid northeast trade winds combined with the altitude and northwest dry winds produce an inversion layer originating a laurel forest with a great floristic diversity.

Fuente de la Canaria Cave is located in Villa de Mazo, southeast of La Palma Island (Fig. 1). The cave, with a length of 237 m, is located a few hundred meters in a southwest direction of the Vinijore mountain, close to La Sabina neighborhood (Fig. S1), at an altitude of 700 m above sea level (U.T.M. 28RBS267654) (Dumpiérrez et al., 1997).

Figure 1 Map of Fuente de la Canaria Cave, La Palma Island.

Planimetry and cross-section of Fuente de la Canaria Cave.

In La Palma Island, forests have been naturally affected by wildfires for centuries and torrential rains after the fires were frequent. In the period 2000–2017, 343 wildfires affected this island. On August 4, 2012 a fire in the municipality of Villa de Mazo destroyed 2,028 Ha, and affected Fuente de la Canaria surface area (Fig. S1), resulting in extensive forest devastation and soil erosion. Transport of soil organic matter and ashes into caves were favored by the low thickness of the topsoil and the porosity of the basalts (Miller et al., 2020a). In addition, the cave shows wide shrinkage cracks and a continuous drip of water from the ceiling, from which the cave owes its name (fountain: fuente).

The entrance to the cave is a jameo (a large opening in the lava tube cave made by the partial collapse of the ceiling) (Fig. 1). This entrance is about 3.5 m long by more than 1 m wide and inside grows a well-developed specimen of the tree heath Erica arborea. Once inside the cave, abundant herbaceous vegetation is noticed in the entrance ground with the presence of the invasive Mexican hygrophile plant Ageratina adenophora (Dumpiérrez et al., 1997). The cave walls present different light to dark brown mucous speleothems and rock walls widely coated by moonmilk and microorganisms (Fig. 2 and Fig. S2).

Figure 2 Samples collected from Fuente de la Canaria Cave.

Field images of the mineral deposits and colored microbial colonies collected from Fuente de la Canaria Cave (MZ03) in La Palma Island (Spain). (A) sample MZ03-2B. (B) sample MZ03-3C. (C) sample MZ03-8H. (D) sample MZ03-10J.

Sampling

Samples were collected in 2015 (MZ03-2B, MZ03-3C, MZ03-8H) and 2016 (MZ03-10J) from the basaltic lava tube cave walls of Fuente de la Canaria Cave in La Palma Island (Canary Islands, Spain) (Fig. 1). Field experiments in La Palma Island caves were approved in project CGL2013-41674-P from the Spanish Ministry of Economy and Competitiveness. The cave temperature was 13.5 °C and the relative humidity 90.5% at the sampling time. Microbial mats and mineral deposits were collected using sterile scalpels and stored in sterile 50 ml tubes. The samples were preserved at 4 °C until arrival at the laboratory and then stored at −80 °C.

A total of four areas were sampled along the cave length, comprising light brown to orange jelly-like secondary mineral deposits developing on the cave walls. Mineral deposits (moonmilk) and microbial colonies were spread all over the wall surface (Fig. 2 and Fig. S2). The characteristics of the samples were as follows: MZ03-2B corresponded to a mucous formation of ochre color; MZ03-3C was an ochre soft stalactite: MZ03-8H an ochre mucous deposit on a rock crack with water runoff; MZ03-10J a mineral formation with abundant moonmilk deposits, extensively represented in the lava tube cave walls.

Field emission scanning electron microscopy

Samples were dried at 50 °C for 24 h, treated and examined in a Jeol JSM-7001F field emission scanning electron microscope as reported by Riquelme et al. (2015b).

DNA extraction, sequencing and phylogenetic analysis

Genomic DNA was extracted from 250 or 500 mg as described elsewhere (Jurado et al., 2020a). The DNA concentration was quantified using a Qubit 2.0 fluorometer (Invitrogen, Carlsbad, CA, USA) in order to reach a minimum of 100 ng/sample.

DNA was analyzed by Next Generation Sequencing. We focused on V3 and V4 regions of the 16S rRNA gene using 341F (CCTACGGGNGGCWGCAG) and 805R (GACTACHVGGGTATCTAATCC) primers (Jurado et al., 2020a). Sequencing was carried out by means of the Illumina MiSeq platform. Amplicon library for samples MZ03-2B, MZ03-3C and MZ03-8H was constructed by Macrogen (Seoul, Korea) for 2 × 250 paired-end sequencing, according to the Illumina metagenomic library preparation protocol, whereas MZ03-10J amplicon library construction was carried out by STAB Vida sequencing services (Portugal) for 2 × 300 paired-end sequencing.

Quality control and trimming of raw data was processed using FASTQC (http://www.bioinformatics.babraham.ac.uk/projects/fastqc/) and Trimmomatic (0.36 version) (Bolger, Lohse & Usadel, 2014), respectively. Paired-end reads were assembled using PEAR (Zhang et al., 2014). QIIME 1.9.1 was used for subsequent analyses (Caporaso et al., 2010). Operational Taxonomic Units (OTUs) were clustered at 97% cutoff using UCLUST (Edgar, 2010). SILVA database for bacteria (version 132) (Quast et al., 2013) for taxonomic identification of 16S rRNA gene sequences (threshold of 80%), heat-maps built in R using gplots package (Warnes et al., 2016), and alpha diversity metrics were used as reported by Jurado et al. (2020a).

Represented samples in heat-maps were reordered with dendrograms based on the row and column mean values as described by the authors. The raw reads were deposited into the NCBI Sequence Read Archive (SRA) database under the accession numbers ERX3225013, ERX3225015, ERX3225016 and SRX9462704.

Ecological function of samples was investigated with the “functional annotation of prokaryotic taxa” software, which employs a manually curated functional annotation database based on the literature of cultured representatives of soil and marine microbiomes (Louca, Parfrey & Doebeli, 2016). A bubble plot illustrating the data was generated using the “ggplot2” library in R package.

The ecological role of microbial comunities was also predicted with a bioinformatics software package (Douglas et al., 2019). This software was used to estimate the funtional profile from the16S rRNA gene obtained data and predict the metabolic-pathways and enzymes involved in nitrogen, sulfur, methane cycles and CO2 fixation. The predictions are given based on MetaCyc database (Karp et al., 2002). A heatmap was created to visualize the abundances of key enzymes using pheatmap package in R.

Assembled genome Ga0077536 from metagenome, accessioned as LNEL00000000, was functionally annotated using Prodigal (Hyatt et al., 2010) for gene prediction, and Sma3s (Muñoz Mérida et al., 2014) along with the curated Uniprot-SwissProt database (UniProt Consortium, 2019) for biological intepretation of genes.

Results and Discussion

Microscopy

FESEM images revealed abundant microbial cells in all the samples. Actinobacterial-like morphologies, similar to those reported by Riquelme et al. (2015a) in other volcanic caves, were observed in the mucous formation (MZ03-2B) (Figs. 3A and 3B). These microbial features comprise coccoid-shaped and filamentous cells with spiny surface ornamentations. Bacterial filaments with smooth surfaces were detected in the ochre stalactite (MZ03-3C), with less than 0.5–1 µm in diameter, probably of Actinobacteria (Fig. 3C). In addition, long hollow mineralized filaments were observed embedded in a glue-like matrix rich in Si (Fig. 3D). The mucous deposit with water runoff (MZ03-8H) showed less abundance of microbial structures. Interestingly, cells with long surface appendages were found in this sample (Fig. 3E), resembling the prosthecate bacterium found in grey vermiculation deposits, reported by D’Angeli et al. (2019) in Fetida Cave, an active sulfuric acid cave in Apulia, Italy. An intricate net of nano-scaled filaments with a collapsed microbial-like structure entrapped in the dense network was also observed in the mucous deposit with water runoff (MZ03-8H) (Fig. 3F). In contrast, microbial cells were abundantly observed in the moonmilk (MZ03-10J), comprising: (i) an interwoven mass of filaments with less than 1 µm in diameter (Fig. 3G), and (ii) clusters of coccoid-shaped cells with smooth surfaces and approximately 1 µm in diameter, which were embedded in extracellular polymeric substances (EPS) and spread all over the sample surface (Fig. 3H).

Figure 3 Field Emission Scanning Electron Microscopy images of samples.

Representative Field Emission Scanning Electron Microscopy images of the studied samples, depicting (A,B) Clusters of Actinobacteria-like cells with spiny surface ornamentation in the mucous formation of ochre color (MZ03-2B); (C) Microbial filaments probably of Actinobacteria in the ochre soft stalactite (MZ03-3C); (D) Hollow bacterial filament on the sample surface of the ochre soft stalactite (MZ03-3C); (E) Prosthecate-like bacterial cell in the ochre mucous deposit on a rock crack with water runoff (MZ03-8H); (F) Dense network of nano-scaled filaments in the ochre mucous deposit on a rock crack with water runoff (MZ03-8H); (G) Mass of interwoven filaments, and (H) Coccoid-shaped cells with smooth surfaces in the mineral formation with abundant moonmilk deposits (MZ03-10J).

Microbial community composition

Alpha diversity metrics were assessed to know both diversity indices and richness estimators of bacterial community in the analyzed samples. On one hand, Simpson 1-D and Shannon-Weaver diversity indices were implemented to provide an insight of the community composition. Simpson 1-D index measures the relative abundance of species, where value of ranges varies between 0 and 1, with samples close to 0 when no diversity is observed and samples with values close to 1 when a high diversity is showed (Simpson, 1949). Shannon-Weaver index is focused on species richness and its value increases when the number of species also increases and the distribution of individuals among species becomes even (Lemos et al., 2011). On the other hand, richness estimators Chao1 and ACE predict the number of species in the sample, focusing on the low abundance of rare species observed (Hughes et al., 2001). Thus, the mucous formation (MZ03-2B) was more diverse and even, whereas the moonmilk (MZ03-10J) showed the lowest diversity. Regarding ACE/Chao1 estimators and the observed species, the mucous deposit with water runoff (MZ03-8H) showed the highest number of species, as would correspond to enrichment from the top soil (Table 1).

Table 1 Indices of microbial community.

Alpha diversity indices for microbial community 16S rRNA amplicon libraries in Fuente de la Canaria Cave.

Samples	Simpson	Shannon	Observed species	ACE	Chao1	
MZ03-3C	0.973	8.736	20511	22006.842	20973.610	
MZ03-8H	0.991	9.459	20882	25515.708	23178.711	
MZ03-2B	0.993	10.063	21191	24162.354	22347.636	
MZ03-10J	0.706	5.033	4650	5752.039	5290.068	

The Venn diagram of shared OTUs among the studied samples resulted in a higher significance between microbial communities from the ochre mucous formation (MZ03-2B) and the mucous deposit with water runoff (MZ03-8H), sharing 6809 OTUs representing the 32.51% and 33%, respectively (Fig. S3). The mucous deposit with water runoff (MZ03-8H) and the ochre stalactite (MZ03-3C) shared the second biggest group of OTUs in common for this study (4021), entailing 19.26% and 19.60%, respectively. However, the ochre stalactite (MZ03-3C) and the mucous deposit with water runoff (MZ03-2B) only shared 6.87% and 6.65% of their total OTUs, respectively. The ochre stalactite (MZ03-3C) showed the highest number of unique groups (52.12%), followed by the moonmilk (MZ03-10J) (47.42%), mucous formation (MZ03-2B) (40.82%), and the mucous deposit with water runoff (MZ03-8H) (26.66%). Thus, the ochre stalactite (MZ03-3C) was reported as the most exclusive of the analyzed samples, whereas the mucous deposit with water runoff (MZ03-8H) pooled the majority of their OTUs with the other samples. All samples shared a total of 788 OTUs, being especially representative for the moonmilk (MZ03-10J), with almost 16.95% of the total OTUs in this sample.

Microbial community structure

A review on the weathering of volcanic (basaltic) rocks revealed that the process has been studied in terrestrial (Kelly et al., 2014) and marine environments (Gulmann et al., 2015). In terrestrial environments (soils) the major groups involved in biological weathering were Actinobacteria, Acidobacteria and Proteobacteria although their abundance varied with the environment and location (Gomez-Alvarez, King & Nüsslein, 2007). Similar phyla distributions were observed in lava tube caves (Riquelme et al., 2015b; Lavoie et al., 2017).

Fuente de la Canaria Cave is affected by the leaching of organic compounds and ashes through cracks and fissures of the volcanic rock (Fig. 2C) as well as the dripping waters from the ceiling. Obviously this must influence the microbial communities. The four samples collected in the cave show an almost exclusive dominance of Bacteria over Archaea and a negligible presence of unassigned prokaryotes (Table 2). Table 3 displays the distribution in phyla and reveals a majority abundance of Proteobacteria (range 89.4–37.6%), followed by Actinobacteria (20.2–3.4%), Acidobacteria (12.6–2.4%) and Candidatus Rokubacteria (8.4–0.4%). These four phyla comprise abundances between 95.8% (moonmilk) and 71.9% (mucous formation).

Table 2 Prokariotic distribution in the cave.

Prokaryotic OTUs distribution in Fuente de la Canaria Cave.

Domain	MZ03-2B	MZ03-3C	MZ03-8H	MZ03-10J	
Archaea	0.339	0.179	0.303	0.036	
Bacteria	99.789	99.805	99.641	99.959	
Unassigned	0.073	0.016	0.056	0.005	

Table 3 Major bacterial phyla in the cave.

Major bacterial phyla in samples from Fuente de la Canaria Cave.

Phylum	MZ03-2B	MZ03-3C	MZ03-8H	MZ03-10J	
Acidobacteria	12.557	7.684	12.260	2.486	
Actinobacteria	13.259	20.236	14.292	3.452	
Bacteroidetes	2.971	0.377	1.897	0.312	
Chloroflexi	3.896	3.370	4.169	0.838	
Candidatus Dadabacteria	1.916	0.468	1.760	0.081	
Candidatus GAL15	1.358	0.486	1.607	0.111	
Gemmatimonadetes	5.018	1.006	4.795	0.277	
Nitrospirae	4.037	2.301	3.238	1.302	
Patescibacteria	1.255	0.521	0.894	0.031	
Planctomycetes	2.835	3.982	3.141	0.635	
Proteobacteria	37.635	53.342	40.101	89.451	
Candidatus Rokubacteria	8.415	3.400	7.594	0.416	
Verrucomicrobia	1.403	0.182	0.941	0.041	
Others (<1%)	3.234	2.450	2.952	0.526	

Comparing the phyla abundance in all the four samples (Table 3), three of them showed relatively similar phyla distribution and abundances regarding the most noticeable phyla (Proteobacteria, Actinobacteria, Acidobacteria and Candidatus Rokubacteria). However, the moonmilk formation, showed about 90% of abundance of Proteobacteria, with minority representation of the other three phyla.

The striking abundance of Proteobacteria is a common feature in lava tube caves from New Mexico, Hawai’i and Azores (Northup et al., 2011; Riquelme et al., 2015b). In addition, the Proteobacteria classes are well represented in caves all over the world (Saiz-Jimenez, 2015), but Gammaproteobacteria are predominant in show caves (e.g., Lascaux Cave, France) impacted by tourism or anthropic activities (Bastian, Alabouvette & Saiz-Jimenez, 2009).

Phyla with abundances below 5% in any of the samples and in decreasing order of abundance were Chloroflexi, Gemmatimonadetes, Nitrospirae, Planctomycetes, Bacteroidetes, Candidatus Dadabacteria, Candidatus GAL15, Verrucomicrobia and Patescibacteria. Other phyla with relative abundances <1% were also retrieved.

At the class taxonomic level (Fig. 4), the heat-map revealed the dominance of Gammaproteobacteria over all other classes, followed by the phylum Actinobacteria. The relative abundance of the different classes of Proteobacteria varied from each sample. The moonmilk showed an abundance of Gammaproteobacteria 2 to 3 times higher than the other three samples. Alphaproteobacteria and Deltaproteobacteria were considerably less abundant. These variances are in agreement with the findings of Gomez-Alvarez, King & Nüsslein (2007) who suggested that differences in the local environment and elemental composition of the volcanic deposits themselves may control bacterial community composition.

Figure 4 Taxonomic identifications of Bacteria at class level.

Heat-map analysis of Fuente de la Canaria Cave with taxonomic identifications of Bacteria at class level. The classes are described in the right column and their respective abundances included in the boxes. Colored left bar groups the classification at phylum level.

Within the Actinobacteria phylum, the classes Acidimicrobiia, Actinobacteria and Nitriliruptoria were represented, but only Actinobacteria was dominant in all samples (13.5–7.2%) except in the moonmilk with 2.8% of relative abundance. Acidimicrobiia ranged between 5.1 and 2.7% in three samples but was insignificant in the moonmilk (MZ03-10J) (0.2%). Nitriliruptoria reached importance in the ochre stalactite with 3% of relative abundance. These three classes are well represented in other lava tube caves and limestone caves worldwide (Riquelme et al., 2015a; Saiz-Jimenez, 2015; Lavoie et al., 2017; Gonzalez-Pimentel et al., 2018).

The class NC10 of Candidatus Rokubacteria was the third group in abundance. In this group, methane oxidation under anaerobic conditions is associated with nitrite reduction (Lomakina et al., 2020) and the relative abundances were important in three samples, and lower in the ochre stalactite MZ03-3C.

The Acidobacteria classes identified in the samples from Fuente de la Canaria Cave comprise Acidobacteriia, Blastocatellia, Holophagae and subdivisions 6 and 17. Blastocatellia was the most abundant class among Acidobacteria, with relative abundances ranging from 4.1% to 1.4%. Acidobacteriia and subdivision 17 ranged between 3.4% and 2.5% in the mucous formation and the mucous deposit with water runoff. A survey on different lava caves from La Palma Island provided evidence of the presence of Acidobacteria, but with relative abundances below 5% (Gonzalez-Pimentel, 2019). Blastocatellia and subdivision 6 were abundant in Altamira Cave (Zimmermann et al., 2005) and in soil (Janssen, 2006) and were reported in coralloids from another La Palma Island cave (Gonzalez-Pimentel et al., 2018). In addition, clones closely related to the genera Luteitalea and Vicinamibacter, from subdivision 6, were retrieved in lava tube caves from Galapagos Islands (Miller et al., 2020b).

The phyla with lower relative abundances, such as Chloroflexi, Gemmatimonadetes, Planctomycetes, Bacteroidetes, Nitrospirae and Verrucomicrobia are commonly found at low rates in lava tube caves (Northup et al., 2011; Riquelme et al., 2015b; Lavoie et al., 2017). Other less common phyla (Candidatus GAL15, Candidatus Dadabacteria) are rarely found. Ca. GAL15 was recovered in Pindal Cave, Spain (unpublished results) and Ca. Dadabacteria was observed in Nerja Cave, Spain (Jurado et al., 2020b).

Figure 5 shows the heat-map of the families and genera retrieved in the four samples from Fuente de la Canaria Cave. The class Gammaproteobacteria is composed of the orders Betaproteobacteriales, Ga0077536, Nitrosococcales, PLTA13 and Steroidobacterales.

Figure 5 Taxonomic identifications of Bacteria at family/genus level.

Heat-map analysis of Fuente de la Canaria Cave with taxonomic identifications of Bacteria at family/genus level. The families/genera are described in the right column and their respective abundances included in the boxes. Colored left bar groups the classification at order level.

The most abundant group belongs to an uncultured bacterium from the gammaproteobacterial order Ga0077536 with 70.5% of relative abundance in the moonmilk, and abundances ranging between 15.4 and 4.6% in the rest of samples. For the order Ga0077536 very short information is available. This lineage has been previously found associated with marine organisms such as corals (Silva-Lima et al., 2020). Functional analysis carried out on the genome assembled from a metagenomic sample (Pinto et al., 2015) resulted in the presence of genes likely involved in sulfur, nitrogen and carbon metabolism. Thus, from 1,278 annotated genes, using Uniprot Swiss-Prot database, out of 4368 totals predicted by Prodigal, 18 genes were related to pathways involved in nitrogen metabolism and 23 to nitrogen fixation, as well as 17 and 9 genes in one-carbon and sulfur metabolism, respectively. This bacterium depicted the whole gene cluster involved in the subpathway that synthesizes formate from formaldehyde [H(4)MPT route], which was formerly described in the methylotrophic bacterium Methylorubrum extorquens (Vorholt et al., 1998). Thus, the high representativeness of Ga0077536, especially in the moonmilk, could play a relevant role in the methane cycle.

Another outstanding group within the Gammaproteobacteria was the order Betaproteobacteriales which encompasses four families, B1-7BS, Burkholderiaceae, Nitrosomonadaceae and a family of uncultured bacterium. This last family was also relatively important in the mucous formation (9.4%) and the mucous deposit with water runoff (5.2%). In the other two samples the abundances were insignificant, below 0.2%. Members of the family B1-7BS were previously retrieved from an Alpine cave (Jurado et al., 2020a) and from a sulfide mineral deposit in USA (Jones et al., 2017). In the family Burkholderiaceae was found the genus Polaromonas, with 2.6% in the mucous formation and 1.7% in the mucous deposit with water runoff.

In Fuente de la Canaria Cave, the family Nitrosomonadaceae comprises the genera IS-44 and MND1. IS-44 was previously found in soils (Zarraonaindia et al., 2015; Heo et al., 2020), and MND1 in caves and soils (Jones, Lyon & Macalady, 2008; Rummel et al., 2020).

The order Nitrosococcales and family Nitrosococcaceae is signified by the genus wb1-P19, with abundances from 13.9% in the ochre stalactite to 1.1% in the moonmilk. This genus was the most abundant group in the vermiculations of an Alpine cave (Jurado et al., 2020a). Other records at lower abundances include caves in different continents (Holmes et al., 2001; Zhu et al., 2019; Anda et al., 2020).

Members of the order PLTA13, which attained some importance only in the ochre stalactite (4.0%) and the mucous deposit with water runoff (1.7%), have been described in soils (Rummel et al., 2020) and in a Swedish mine of rare earth elements (Sjöberg et al., 2017).

The genus Steroidobacter was identified within the order Steroidobacterales with about 10% of relative abundance in the moonmilk. This genus was moderately common in limestone and volcanic caves (Porca et al., 2012; Riquelme et al., 2015b; Lavoie et al., 2017).

The order Rhizobiales (Alphaproteobacteria) included four families from which the Beijerinckiaceae with an uncultured bacterium had relative abundances around 2–3% in all the samples, except in the moonmilk with 0.3%. The only identified genus was Hyphomicrobium, with relative abundances of 1.7% in the moonmilk and below 1% in the other samples.

The actinobacterial family Pseudonocardiaceae was represented by the genus Crossiella, with relative abundances above 12% in the ochre stalactite and 6% in the mucous formation and the mucous deposit with water runoff. In the moonmilk this genus was relatively minor, about 2.5% of abundance. Crossiella is a dominant member of the microbial communities of lava tube caves (Riquelme et al., 2015b; Spilde et al., 2016) and its moonmilk deposits (Miller et al., 2020b). The family Euzebyaceae only reached some importance in the ochre stalactite. Euzebyaceae was very abundant in microbial mats on coralloids from a lava tube cave of La Palma Island (Gonzalez-Pimentel et al., 2018) and Euzebyales was the second most abundant order (in number of sequences) in New Mexico and Hawai’i lava tube caves (Riquelme et al., 2015a).

The classes Acidimicrobiia and Actinobacteria contain uncultured bacteria. Metagenomic analyses have revealed that there were many uncultured actinobacterial species belonging to the class Acidimicrobiia in subterranean environments and acid mine drainage, in addition to freshwater and marine samples (Gonzalez-Pimentel, 2019).

Whitin the phylum Acidobacteria relative abundances above 2% only were found in the mucous formation and mucous deposit with water runoff for Acidobacteriia, and for the clade 11-24 in the ochre stalactite. Both 11-24 and RB41 clades, in addition to Steroidobacter and Nitrospira, all of them were retrieved in this study and identified as members of the rhizosphere bacterial community (Liu, Jin & Guo, 2020). The uncultivated clade RB41 was very abundant in tundra soils (Ivanova et al., 2020). These findings are in agreement with the presence of roots in the shallow cave ceiling.

The genus Nitrospira, within the family Nitrospiraceae, as well as Candidatus Methylomirabilis (Rokubacteria) attained abundances over 2% in most samples. Nitrospira comprises ammonia-oxidizing bacteria which are relatively common in caves (Tomczyk-Zak & Zielenkiewicz, 2016). Nitrospira was previously found in volcanic caves from California, New Mexico, Hawai’i and Azores (Northup et al., 2011; Riquelme et al., 2015b; Lavoie et al., 2017). Candidatus Methylomirabilis is a denitrifying methanotroph (Wu et al., 2012). The clade wb1-A12 was retrieved from an Australian cave (Holmes et al., 2001). Therefore, it appears that nitrifiers are widespread in the cave.

Gemmatimonadetes are relatively common in lava tube caves and limestone caves (Northup et al., 2011; Saiz-Jimenez, 2015). DeBruyn et al. (2011) reported that bacteria belonging to the phylum Gemmatimonadetes comprise approximately 2% of soil bacterial communities, have a generalist ecological strategy and adapt to a variety of environments. Gemmatimonadaceae were well represented (around 5%) in the mucous formation and the mucous deposit with water runoff.

In Fuente de la Canaria Cave Planctomycetacia members were affiliated with the family Gemmataceae, and Bacteroidetes with the order Kryptoniales (family BSV26), both with low relative abundances. This last family has been retrieved from shallow groundwater aquifer in southeastern Wisconsin (Gayner, 2018) and from Florida lagoon sediments (Bradshaw, 2020).

Candidatus Rokubacteria, Gemmatimonadetes, Candidatus Dadabacteria, Patescibacteria, Verrucomicrobia, Candidatus GAL15, and some others phyla are part of a so-called rare cave biosphere (Hershey & Barton, 2018). Most of these phyla have been retrieved from soils and subsurface soils (Hug et al., 2016; Zeng et al., 2016; Brewer et al., 2019; Nixon et al., 2019; Lemos et al., 2020; Sharrar et al., 2020).

It must be noticed the close similarity in taxa and relative abundances of both mucous samples of ochre color (MZ03-2B and MZ03-8H), although the last was located on a rock crack with water runoff. The sample MZ03-10J, with abundant moonmilk deposits, very different from the others, represented a niche almost completely dominated by Proteobacteria.

Regarding the participation of the bacteria in the geochemical cycle of elements, some insights can be derived from the taxonomical groups retrieved in Fuente de la Canaria Cave. The ecological functions of the bacterial communities were analyzed by FAPROTAX (Fig. 6). A total of 54 microbial functional groups corresponding to 19% of OTUs were identified. Within the identified OTUs we found a high proportion assigned to aerobic chemoheterotrophy, nitrification and methanotrophy across all the samples.

Figure 6 Predicted ecological functions of Bacteria.

Relative abundance of FAPROTAX predicted ecological functions (Y axis) of Fuente de la Canaria Cave. The size of the cycles indicates the relative abundance.

The major predicted ecological function was chemoheterotrophy, in which were involved the phyla Proteobacteria (Alpha- and Gammaproteobacteria) and Actinobacteria. Crossiella was the most representative genera in this function, in addition to other numerous genera, with lower relative abundance.

Microbial community functional profiles of samples collected in Fuente de la Canaria Cave were also predicted using PICRUSt2 software. In this study we focused on the enzymes involved in methane, sulfur, nitrogen metabolism and CO2 fixation (Fig. 7).

Figure 7 Predicted genes encoding enzymes.

Heat-map showing the relative abundances of PICRUSt2 predicted genes (Y axis) encoding the enzymes involved in methane, sulfur, nitrogen metabolism and CO2 fixation based on Metacyc database for each sampling point (X axis). The values for functional groups are marked by colors from white to red designating the least abundant to most abundant.

The predictive accuracy of PICRUSt2 was evaluated by the weighted Nearest Sequenced Taxon Index (NSTI score), which reflects how similar are the microorganisms in a given sample when comparing with available reference genomes. According to developers, PICRUSt produces accurate metagenome predictions with a mean NSTI of 0.17. The mean NSTI score of the sequences was 0.20 for the ochre stalactite and 0.21 for the other three samples, close to those reported for other environmental recent studies with higher mean NSTI values (Martin-Pozas et al., 2020). The unexplored diversity in complex environmental communities was the main cause of the lower observed degree of accuracy and the results should be treated with caution. However the predictions could still provide some important insights into the microbial ecological functions of underexplored environments such as lava tube caves.

Methane metabolism

Two genera from Fuente de la Canaria Cave were involved in methylotrophy, Candidatus Methylomirabilis, a denitrifying methanotroph (Wu et al., 2012) and the genus Hyphomicrobium that can grow with low concentrations of C1 compounds such as methanol, methylamine, and others (Oren & Xu, 2014). However, Hyphomicrobium was not confirmed by FAPROTAX. Other possible methanotrophic bacteria have been identified within the family Beijerinckiaceae. This family comprises obligate methanotrophs and examples of the intermediate states: facultative methylotrophs and facultative methanotrophs, in addition to chemoorganoheterotrophs. This metabolic trait was confirmed by FAPROTAX (Fig. 6). The functional profiles of the ochre deposit samples (MZ03-2B and MZ03-8H) with respect to the stalactite and moonmilk samples (MZ03-3C and MZ03-10J) presented an increasing level of methanotrophy, methylotrophy and hydrocarbon degradation functional groups which were associated with the families Beijerinckiaceae, Methylococcaceae and Methylomirabilaceae (Fig. 5). A deeper analysis of key methane metabolism enzymes (Fig. 7) revealed an increase of the particulate methane monoxygenase in the three ochre samples, which is mainly associated with the family Beijerinckiaceae. In addition, PICRUSt2 metabolic predictions revealed Hyphomicrobium, MND1 and IS-44 and Nitrospira encoded enzymes with the potential to support catabolic methane or ammonia oxidation, as genes encoding particulate methane monooxygenase and ammonia monooxygenase share high sequence identity (Holmes et al., 1995). In general, the genes related to methane metabolism presented a low abundance in the moonmilk. This could be due to the absence of metabolic information on Ga0077536 bacterium in PICRUSt2 database, since gene prediction and annotation analysis on the genome resulted in the identification of the putative dehydrogenase XoxF (EC: 1.1.2.7), the formylmethanofuran–tetrahydromethanopterin formyltransferase (EC: 2.3.1.101), the 5,6,7,8-tetrahydromethanopterin hydro-lyase (E.C: 4.2.1.147), the methenyltetrahydromethanopterin cyclohydrolase (E.C: 3.5.4.27), the aralkylamine dehydrogenase light chain (E.C: 1.4.9.1), all these enzymes found unequally in the analyzed samples. Beyond these proteins, the presence of putative glutathione-dependent formaldehyde-activating enzyme (E.C: 4.4.1.22) was observed in this bacterium, but not in the rest of bacteria.

A metagenomic study of the gammaproteobacterial Ga0077536 showed the putative presence of glutathione-dependent formaldehyde-activating gene, which is related to methane metabolism. For methylotrophic bacteria, formaldehyde is also a central intermediate for oxidizing methanol or methylamine (Simon et al., 2017).

There is previous evidence of microbial methane oxidation in caves. Fernandez-Cortes et al. (2015) investigated methane consumption in several Spanish caves and showed the presence of Methylocapsa aurea, Methylomicrobium album, Methylococcus capsulatus and K1-1 and K3-16 methanotrophs in well-ventilated caves. Webster et al. (2018) studied 42 sediment samples from 21 caves in North America and found methanotrophs in 88% of the samples. Pratscher et al. (2018) reported the presence of microbial groups containing the upland soil cluster α, responsible for most of the methane uptake, in volcanic soils and in volcanic cave wall biofilms all over the world. Martin-Pozas et al. (2020) reviewed the role of methane-oxidizing bacteria in caves. These bacteria not only consume large amounts of the methane that enters to the underground atmosphere but also produce bioactive compounds, such as methanobactins, with potential application in medicine.

Sulfur metabolism

Proteobacteria contains sulfur oxidizing and reducing lineages. In fact, Fig. 6 shows evidence of sulfur and sulfate respiration and in the moonmilk was associated to the order Ga0077536 (Table S2). In particular, the genes cysI and cysJ, known to be involved in the subpathway that synthesizes hydrogen sulfide from sulfite (NADPH route), were predicted as alpha and beta subunits of sulfite reductase activity.

CO2 fixation

The gene families related to CO2 fixation were represented across all samples. The most abundant CO2 fixation pathways predicted by PICRUSt2 were Calvin-Benson-Bassham (CBB) cycle and the reductive tricarboxylic acid (TCA) cycle. As many enzymes involved in these pathways are involved in other multiple pathways, we examined the presence of key enzymes of the known autotrophic CO2 fixation pathways. Ribulose-biphosphate carboxylase/oxygenase (RubisCo), responsible for CO2 fixation, is the key enzyme of the CBB cycle, which is not only extensive to phototrophic environments but has been described in chemoautotrophic proteobacteria that perform ‘dark’ CO2 fixation in diverse habitats (Tsai et al., 2015). The phosphoribulokinase (prkB) enzyme rarely occurs in organisms that lack the CBB cycle (Frolov et al., 2019). Both enzymes were mainly predicted in the family Nitrosomonadaceae, some uncultured members from the gammaproteobacterial order Ga0077536, and the genera Hyphomicrobium and Polaromonas (Table S3). Most of the enzymes of the TCA cycle could catalyze both directions, but the enzymes that catalyzed the reductive direction (ATP citrate lyase, 2-oxoglutarate:ferredoxin oxidoreductase, pyruvate:ferredoxin oxidoreductase and fumarate reductase) were only associated with the genus Nitrospira. No other CO2 fixation pathways, such as the reductive acetyl coenzyme A pathway, and the 3-hydroxypropionate cycle, could be found in PICRUSt2; however, it is important to emphasize the identification of the subunits alpha and beta of the acetyl-coenzyme A carboxylase carboxyl transferase (E.C: 2.1.3.15) in Ga0077536 bacterium, since these enzymes could be relevant in CO2 fixation process in prokaryotes (Hügler et al., 2003). First, biotin carboxylase (E.C: 6.3.4.14) catalyzes the carboxylation of biotin on its carrier protein (BCCP) and then the CO2 group is transferred by the carboxyl transferase to acetyl-CoA to form malonyl-CoA. To summarize, gammaproteobacterial order Ga0077536 was very abundant in moonmilk deposits and likely involved in calcite deposition.

Nitrogen metabolism

The nitrogen cycle is represented by the family Nitrosomonadaceae, all of whose cultivated representatives are lithoautotrophic ammonia oxidizers (Prosser, Head & Stein, 2014) and the genus Nitrospira composed of ubiquitous nitrite-oxidizing bacteria (Koch et al., 2015). Indeed, nitrogen fixation is a remarkable trait of the family Beijerinckiaceae (Marín & Ruiz Arahal, 2014) and several members of the family Burkholderiaceae (Coenye, 2014). This allows these microorganisms to thrive in niches of scarce nitrogen availability, as in some oligotrophic caves. FAPROTAX assigned high relative abundances to aerobic ammonia oxidation (Nitrosomonadaceae and Candidatus Nitrososphaera) and aerobic nitrite oxidation (Nitrospira, Leptospirillum and Nitrosomonadaceae). This metabolic trait was also predicted by PICRUSt2. Relatively lower were other nitrogen-related activities, such as denitrification, nitrate respiration, nitrate reduction, or nitrogen respiration.

Interestingly, the abundance of the enzymes hydroxylamine reductase and nitrogenase was high in the ochre stalactite, followed by the moonmilk (Table S4). The first enzyme was mainly related to an uncultured bacterium from Ga0077536. However the presence of the enzyme nitrogenase was predicted for uncultured members from Beijerinckiaceae, an uncultured bacterium from Ga0077536, and uncultured members of the genus Crossiella. The results suggested that white precipitations and ochre biofilms might be important sites for nitrogen cycling in caves, particularly nitrogen fixation.

Of interest is the abundance of the enzyme urease in the moonmilk, also present in the rest of samples, with lower relative abundance. PICRUSt2 analyses reported that urease activity in the moonmilk was associated to an uncultured bacterium from the order Ga0077536. On the contrary, in the other samples, the urease activity was related to the presence of Crossiella and uncultured members within the family Beijerinckiaceae (Table S4).

The ochre stalactite was characterized by large quantities of Crossiella; this genus is a common dweller of lava tube caves of Hawai’i and Azores (Riquelme et al., 2015a; Riquelme et al., 2015b; Spilde et al., 2016) and also in limestone caves (Wiseschart et al., 2019), although at lower abundances than in lava tube caves. Gonzalez-Pimentel (2019) found an extraordinary abundance of Crossiella in moonmilk from La Palma lava tube caves.

Cuezva et al. (2020) reported that Crossiella found in moonmilk have the ability to capture CO2 from the underground atmosphere, resulting in precipitation of calcium carbonate as a by-product of the action of carbonic anhydrase. Sanchez-Moral et al. (2012) reported that in the early stages of moonmilk deposition bacteria induces carbonate precipitation, but subsequently a microbial deactivation occurs when carbonate accumulates.

For Li et al. (2018) Crossiella, likely the primary cause of CaCO3 precipitation, was the dominant genus (about 84% of relative abundance) in a sample of white stains collected from the surface of a statue inside a Chinese cave. This sample presented aggregates of microorganisms and inorganic minerals and the analysis by SEM-EDS revealed that the stains were mostly composed of calcium carbonate.

From the review of all available literature it becomes clear that Crossiella is a common inhabitant of caves and colonizes moonmilk deposits, where it must have an important role in calcite precipitation.

Crossiella is able to hydrolyze or decompose urea (Franco & Labeda, 2014). Ureolytic bacteria are associated with high rates of calcium carbonate precipitation in alkaline environments rich in Ca2+ ions, such as caves. Several authors have reported calcium carbonate precipitation by ureolytic bacteria from caves (Omoregie, Ong & Nissom, 2018; Enyedi et al., 2020). This strongly suggests a principal role of Crossiella in microbially-induced carbonate precipitation (MICP) in caves, and particularly in moonmilk deposits.

However, not only ureolysis is involved in MICP. A few authors reported that other five metabolic pathways can induce this precipitation: photosynthesis, ammonification, denitrification, sulfate reduction, and methane oxidation (Zhu & Dittrich, 2016; Omoregie, Ong & Nissom, 2018). Most of these processes were predicted by FAPROTAX in Fuente de la Canaria Cave. Moreover, cell walls and extracellular polymeric substances can serve as templates for carbonate precipitation (Enyedi et al., 2020).

Other elements

Other mineral elements in the volcanic rock from Fuente de la Canaria Cave are iron and manganese (Carracedo et al., 2001). With abundances below 2% were identified two genera: Hyphomicrobium and Polaromonas. Hyphomicrobium, a prosthecate bacterium (Oren & Xu, 2014) (Fig. 2E), is relatively frequent in cave ferromanganese deposits (Northup et al., 2003; Spilde, Northup & Boston, 2006). This genus is known to mediate the oxidation and precipitation of manganese and iron in different environments (Ghiorse & Hirsch, 1979). The genus Hyphomicrobium and the lineage wb1-A12 were also related with manganese nodules (Molari et al., 2020). Polaromonas was found in manganese deposits from an Italian cave (Vaccarelli et al., 2021).

Conclusions

Different types of speleothems and colored microbial mats, from Fuente de la Canaria Cave, a representative lava tube cave, in La Palma Island, Spain, were studied using DNA next-generation sequencing.

The distribution in phyla revealed a majority abundance of Proteobacteria, followed by Actinobacteria, Acidobacteria and Candidatus Rokubacteria. These four phyla comprised a total relative abundance of 72–96%. The main ecological functions were chemoheterotrophy, methanotrophy, sulfur and nitrogen metabolisms and CO2 fixation; although a wide diversity of other ecological functions was outlined. The abundant presence of lineage Ga0077536, in an extensive area coated by moonmilk, points to a further research in Fuente de la Canaria Cave aiming at the isolation of this bacterium.

The detection of several putative lineages associated with C, N, S, Fe and Mn cycling indicated that Fuente de la Canaria Cave basalt surfaces were colonized by metabolically diverse prokaryotic communities involved in the biogeochemical cycling of major elements.

The microbial communities of lava tube caves from the volcanic Canarian, Azorean and Hawaiian Islands are comparatively similar regarding the distribution of major phyla. Although different geographical locations and environmental conditions can contribute to the diversity and abundance of minor phyla, the data suggest that the volcanic rocks largely determine the distribution of the microbial communities of lava tube caves. This distribution is also governed by natural and/or anthropogenic conditions of the overlying surface layers.

Supplemental Information

Supplemental Information 1 Supplemental Information on Fuente de la Canaria Cave

Click here for additional data file.

Additional Information and Declarations

Competing Interests

Author Contributions

Field Study Permissions

Data Availability

Octavio Fernandez-Lorenzo is the president of Grupo de Espeleologia Tebexcorade, a non-profit sports club of speleologists.

Jose Luis Gonzalez-Pimentel, Tamara Martin-Pozas analyzed the data, prepared figures and/or tables, authored or reviewed drafts of the paper, and approved the final draft.

Valme Jurado and Ana Zelia Miller performed the experiments, analyzed the data, prepared figures and/or tables, authored or reviewed drafts of the paper, and approved the final draft.

Ana Teresa Caldeira analyzed the data, authored or reviewed drafts of the paper, and approved the final draft.

Octavio Fernandez-Lorenzo performed the experiments, analyzed the data, prepared figures and/or tables, and approved the final draft.

Sergio Sanchez-Moral conceived and designed the experiments, analyzed the data, prepared figures and/or tables, authored or reviewed drafts of the paper, and approved the final draft.

Cesareo Saiz-Jimenez conceived and designed the experiments, analyzed the data, authored or reviewed drafts of the paper, and approved the final draft.

The following information was supplied relating to field study approvals (i.e., approving body and any reference numbers):

Field experiments were approved by the Spanish Ministry of Economy and Competitiveness (project CGL2013-41674-P) in La Palma Island caves.

The following information was supplied regarding data availability:

The raw reads are available in the NCBI Sequence Read Archive (SRA) database: ERX3225013, ERX3225015, ERX3225016 and SRX9462704.

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
