# Peer review of "Prokaryotic communities from a lava tube cave in La Palma Island (Spain) are involved in the biogeochemical cycle of major elements"

_PeerJ, doi:10.7717/peerj.11386_

## Round 0.1 · original submission · Minor Revisions

Please address critiques of the reviewers and amend manuscript accordingly.

·

Basic reporting

English is clear and unambiguous throughout the manuscript. The references are appropriate and sufficient. The structure of the article, the figures, and the tables are fine. The results are relevant.

Experimental design

No comment

Validity of the findings

No comments

Additional comments

I enjoyed reading the manuscript ” Prokaryotic communities from a lava tube in La Palma Island (Spain) are involved in carbon fixation, methanotrophy and the biogeochemical cycle of major elements “.
This is a very thorough and comprehensive analysis of the microbial communities isolated at four sites of a lava tube in the island of Las Palmas in the Canary Islands. The samples were sequenced by NGS through the Illumina platform and were also analyzed by FAPROTAX software and PICRUSt2 to have an overview of the trophism of the microbial community also at the level of abundance of the single sequences. The enzymes associated with nitrogen, sulfur, methane metabolism and CO2 fixation were analyzed to construct an overview of community trophism.
I would like to ask the authors a question: are the four microbial communities independent from the outside or do they depend, at least in part, on external trophic inputs, in particular with regard to the carbon source.?

The work is certainly interesting and worthy of being published.

Line 52: change “lava tubes with those Mars” with “lava tubes with those of Mars”

·

Basic reporting

The manuscript intitled "Prokaryotic communities from a lava tube in La Palma Island (Spain) are involved in carbon fixation, methanotrophy and the biogeochemical cycle of major elements" is well written and clear, in my opinion no English revision is required.
Literature references are appropriate. I suggest to check if all of them are in Italic in the text (such as refs in line 413 that needs to be corrected).
Structure is in line with the requirement of the journal.

Experimental design

Experimental design is correct. The aim of the research well defined

Validity of the findings

Findings are relevant because the lava tubes are a not well studied and conclusions are well stated

Additional comments

Well done manuscript!

·

Basic reporting

Your title is misleading. You mostly found chemoheterotrophs.

Overall the English is good and clear, but it is not sophisticated. It needs some copy editing. I have done some.

You keep using your sample code to discuss the results, like sample MZ03-10J, from a moonmilk formation rather than using the direct descriptor and calling it moonmilk. I suggest using the descriptor to make it easier for the reader than an obtuse code. Or both.

You keep being amazed that your moonmilk sample is different than the others. I’m not. I think you should look up some more moonmilk references. Portillo MC, Gonzalez JM. Moonmilk deposits originate from specific bacterial communities in Altamira Cave (Spain). Microb Ecol. 2011 Jan;61(1):182-9. doi: 10.1007/s00248-010-9731-5.

Abstract
"22-23 Lava tubes are particular caves due to their genesis and mineral composition different from
24 karstic caves."
Lava caves differ from karstic caves in their genesis and mineral composition. [lava tubes is a term that is a bit out of date due to changes in our understanding of how lava caves form. Suggest changing tube to cave. I do sometimes use lave tube cave.]

The subsurface microbiology of a lava cave in the Canary Islands…
You say “the most important volcanic archipelago in the Atlantic Ocean”, but what is that based on? You don’t return to that in the introduction. Justify that extreme statement. Refs?
29 comma after profiling
35 functions was were outlined
38 microbially-induced
49 replace the present with this
51 search for recognizable
52 with those of Mars
65 study of Canary
68 from the Canary Islands
68 replace ‘and with this aims’ with ‘so’ we focused
70 replace ‘presenting’ with ‘with’
Results and Discussion
178 0.5 1 m. Is this 0.5-1 m?
181 del; , as revealed by FESEM
189 space needed: substances(EPS)
193 in the analyzed samples
203 del to be discovered
205 del studied
218 process has
224 show
230-235 It is not remarkable that the moonmilk was different. The other three were slimes?
243 Isn’t Dadabacteria still a candidate phylum?
248 Not remarkable. Moonmilk is different.
370 from the others
393 underexplored
423 ‘highlighting on the rest of bacterial community where no presence was evinced.’ I am not sure what you are saying.
431 ‘in caves where methane usually reaches concentrations near to the atmospheric background levels.’ Not clear to me. How can you tell a difference from background levels?
434 responsible for most
436 You say ‘Recently,
436 Martin-Pozas et al. (2020) reviewed the role of methane-oxidizing bacteria in caves.’ Can you briefly state the role?
446 del studied [you have no samples that were not studied]
485 to uncultured bacterium. Should be; an uncultured bacterium, or uncultured bacteria, depending on how many. Also ln 492

Interesting discussion of moonmilk.
508 where it must have
512 karstic caves have lots of calcium carbonate; lava caves not so much, although there is an interesting case in Korea. Jeju volcanic island and lava tubes.
514microbially-induced
526 With abundances below 2%, we identified… [you have discussed Hyphomicrobium a lot in this paper. Why such interest in an uncommon oligotropic bacterium?]
550 ‘as a wide diversity of taxa are even found in different niches or substrata within the same cave.’ I am not quite sure the point you are trying to make. In and of itself it doesn’t make sense to me as your final conclusion, which should be your big insight.
These are mostly copy edits.
References: Really nice job.
PLoS is now PLOS
Jones is out of order
Lomakina. 12(1), 10;
Luis- Vargas. No space before Vargas
Miller and Garcia-Sanchez comes before Miller and Pereira
Omoregic J name not in italics

Riquelme and Hathaway is before Riquelme and Rigal. You’ll also have to change 2015a and 2015b in the refs and in the text

Spilde et al 2016. Name of J not in italics
Vaccarelli. Should you add an English translation of the title?
Wu. Should genus and species be in italics?
Zimmerman should follow Zhu refs.

Experimental design

Overall complete and thorough; meets your expectations. Some additional detail needed in description of the cave. and location and year of sample collection.

Materials and Methods
78 of the Canary Islands
85 in the last years; in recent years
90 So does this amount of rainfall make it a tropical rainforest?
95 in a southwest direction
97 replace ‘being its location’ with ‘located in’
100 jameo has about; replace with jameo is about
100 I don’t understand. Usually entrances are described in terms of width and height. Does length describe the length of the entrance passage? Does not strike me as large.
101 What is Erica arborea. Perhaps add common name.

Is the whole length of the cave 3.5? No. But confusing.
113 stored in sterile
156 del quotations, and ln 160, and 166
“pheatmap” is a new one on me! But I read about it. Nice use of it.

Validity of the findings

Why have the sample numbers been expanded to add another level of numbering? What does that mean in terms of what these samples are?

I think they have done an exceptionally good job of using figures and tables to support their discussion and conclusions.

Figure 1; good map. Show which samples were collected from each location.
Fig 2. Good, but delete ‘sample’ from in front of the sample numbers, and add a brief description. Should MX08 have a hyphen like the other numbers?
Fig 3. E and F are so dark you can’t make out any details.
Fig. 4. Very nice. Add sample descriptor to sample numbers. i.e. moonmilk
Fig. 5. Is order the most appropriate level of classification? Less familiar to most readers, I think.
Fig. 6. Add sample descriptors to sample numbers. Perhaps invert so that the most common functions are at the top?
Fig. 7. Add sample descriptor to sample numbers.
Table 1. Add sample descriptor to sample numbers.
Same for other tables.

Supplemental
SF1; Fire. What fire? Should this info be included in the paper? Why is it important, supplemental or not? Is it relevant to the current study?
SF2: add sample descriptors.

Additional comments

Nothing to add. I am impressed by the amount of information you had to work through with clarity and thoughtfulness.

---

## Round 0.2 · accepted · Accept

All concerns of the reviewers are adequately addressed and the manuscript is amended accordingly. Therefore, the revised version is acceptable now.